# Graves' disease diagnosed in remnant thyroid after lobectomy for thyroid cancer

**Meihua Jin**◉**, Ahreum Jang, Won Gu Kim, Tae Yong Kim, Won Bae Kim, Young Kee Shong, Min Ji Jeon**\*

Divsion of endocrinology and metabolism, Department of Internal Medicine, Asan Medical Center, University of Ulsan College of Medicine, Seoul

\* mj080332@gmail.com

## Abstract

### Background

The coexistence of Graves' disease with thyroid cancer is well-known and total thyroidectomy is recommended in such cases. However, Graves' disease might be dormant at the time of surgery and diagnosed after lobectomy for thyroid cancer.

### Methods

We assessed the incidence and clinicopathological characteristic of newly developed Graves' disease after lobectomy for thyroid cancer between 2010 and 2019.

### Results

In all, 11043 patients underwent lobectomy for thyroid cancer during the study period, and 26 (0.2%) were diagnosed with Graves' disease during follow-up. The median age was 43.8 years, 88.5% were female, and all were euthyroid before surgery. The median time from lobectomy to the diagnosis of Graves' disease was 3.3 years. Half of the patients were diagnosed based on thyroid function tests during routine follow-up, and others were diagnosed due to symptoms of thyrotoxicosis. Among patients who had checked preoperative thyroid autoantibodies, 61.1% showed positivity. Twenty-one (80.8%), and 2 (7.7%) patients received antithyroid drugs and radioactive iodine therapy, respectively, and 3 (11.5%) underwent completion thyroidectomy.

### Conclusion

Although rare, Graves' disease can occur in the remnant thyroid after lobectomy. Such patients are more likely to have autoantibodies. An appropriate workup is required when hyperthyroidism is found during the follow-up of patients after lobectomy.

**Data Availability Statement:** All relevant data are within the manuscript.

**Funding:** This study was supported by a grant (2021IL0025) from the Asan Institute for Life

Sciences, Asan Medical Center, Seoul, Korea for Min Ji Jeon.(https://ails.amc.seoul.kr/ails/en/main/main.do) The funders had no role in study design, data collection and analysis, decision to publish, or preparation of the manuscript.

**Competing interests:** The authors have declared that no competing interests exist.

## Introduction

Graves' disease (GD) is an autoimmune disorder and is considered the most common cause of hyperthyroidism, followed by toxic multinodular goiter and toxic adenoma [1, 2]. Stimulation of the thyrotropin receptor by thyrotropin receptor antibodies (TRAb) is the primary mechanism of GD, which results in the production and release of thyroid hormones [3]. The coexistence of thyroid cancer with GD is well known, and the American Thyroid Association (ATA) states that thyroid cancer occurs in up to 2% of patients with GD [3–5]. In a previous study, among 847 patients with GD who underwent thyroidectomy, the incidence of coexistent thyroid cancer was 4.3%, and 68.2% were papillary microcarcinomas (PTMC) [6]. Although not clearly known, the probable mechanism of increased prevalence of thyroid cancer in patients with GD is primarily the binding of TRAb to thyrotropin receptor, which promotes tumor formation, angiogenesis, and further progression of the invasiveness of cancer [7–9].

Usually, near-total or total thyroidectomy is recommended in patients having thyroid cancer with underlying GD [3]. However, in the absence of underlying GD, thyroid lobectomy is preferred if indicated [10, 11]. The recent ATA guidelines recommended thyroid lobectomy as the initial surgical approach for low-risk PTMCs and for low-risk papillary thyroid carcinoma (PTCs) of size 1–4 cm [10]. As the cases of lobectomy increases, GD diagnosed in the remnant thyroid lobe in some cases has been reported [12–14]. However, the incidence of newly diagnosed GD after thyroid lobectomy is not well known. Furthermore, due to its rarity, little is known about preoperative factors that can predict the development of GD after lobectomy. Therefore, a cohort study with large sample size is necessary to fill this gap.

We aimed to assess the incidence of newly developed GD after lobectomy for thyroid cancer in a retrospective cohort study conducted in a single tertiary center in Korea. Additionally, we aimed to evaluate the clinical and pathological characteristics of these patients and determine factors that might be helpful to predict the occurrence of GD after lobectomy.

## Methods

### Patients

We retrospectively reviewed the medical records of patients who underwent lobectomy for thyroid cancer between 2010 and 2019 at a tertiary medical center in Korea and were diagnosed with GD after thyroid lobectomy (Fig 1). This study was approved by the Institutional Review Board of the Asan Medical Center (No.: 2021–0621).

### Follow-up protocol for thyroid cancer and diagnosis of GD

The data of preoperative thyroid function tests and neck ultrasonography (US) were available for all patients, and tests for preoperative autoantibodies were optionally performed at the discretion of the treating physician. Patients underwent lobectomy with/without prophylactic central neck dissection according to the surgeon's decision. After thyroid surgery, serum free T4 (fT4) and TSH were measured within the first 2 to 3 months, and patients with overt hypothyroidism or subclinical hypothyroidism with a TSH level of >10 mIU/L were treated with levothyroxine [15]. During the long-term follow-up, patients were regularly subjected to physical examination, thyroid function tests, serum thyroglobulin, thyroglobulin antibody (TgAb) level measurements every 6–12 months, and neck US every 12–24 months. GD was suspected when the patient complained of symptoms of thyrotoxicosis, or the thyroid function tests showed overt or subclinical hyperthyroidism during routine follow-up. If the patient was taking levothyroxine, the thyroid function tests were performed again after discontinuation of levothyroxine. Serum thyrotropin-binding inhibitory immunoglobulin (TBII) and

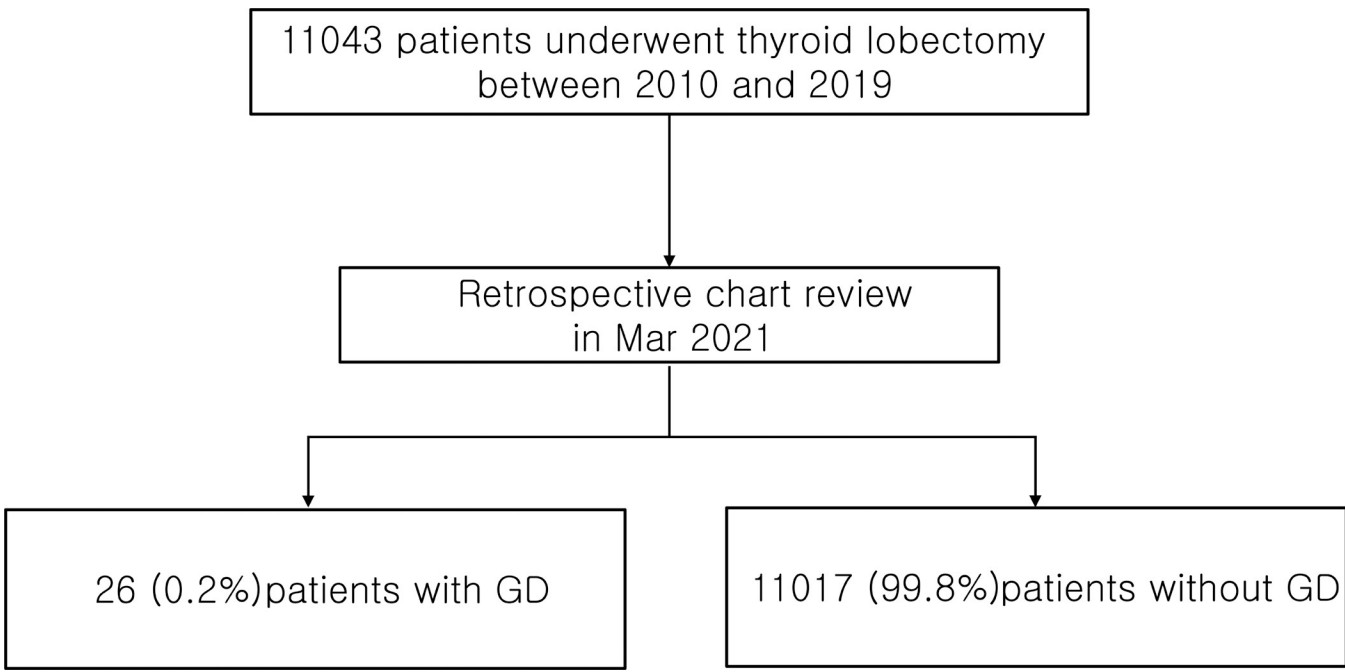

**Fig 1. Flowchart of the patient recruitment for the study.** We found 26 (0.2%) patients who were diagnosed with Graves' disease after lobectomy on retrospective chart review.

$^{99m}$-technetium (Tc) thyroid scan uptake studies were also performed in patients with symptoms of thyrotoxicosis to distinguish from painless thyroiditis. The presence of goiter was determined according to the World Health Organization (WHO) goiter classification system via physical examination by a physician [16]. Thyroid-associated orbitopathy (TAO) was diagnosed either by experienced endocrinologists or ophthalmologists.

### Treatment and follow-up for GD

Patients with GD were initially treated with antithyroid drugs (ATDs); methimazole (15–30 mg/day), or carbimazole (20–40 mg/day). Serum fT4, TSH, and TBII were regularly measured every 2–3 months from the initiation of ATDs. Radioactive Iodine (RAI) therapy and completion thyroidectomy was considered when the patients failed to achieve a euthyroid state despite ATD treatment or if the patient preferred these treatment options.

### Laboratory measurement

Serum TSH levels were measured using the TSH-CTK-3a radioimmunoassay (DiaSorin SpA, Saluggia, Italy) with a functional sensitivity of 0.07 mU/L. Serum fT4 levels were measured by radioimmunoassay (Immunotech, Prague, Czech Republic) with a functional sensitivity of 2.34 pmol/L [17, 18]. The reference ranges of TSH and fT4 were 0.4–4.5 mIU/L and 0.80–1.90 mg/dL, respectively. TBII was measured using the B·R·A·H·M·S TRANK human immunoradiometric assay (B·R·A·H·M·S GmbH, Hennigsdorf /Berlin, Germany) and titer $\geq$1.5 IU/L were considered positive with a functional sensitivity of 1.0 ± 0.2 IU/L. The thyroid peroxidase antibody (TPOAb) level was determined by radioimmunoassay (BRAHMS anti-TPOn RIA), and a value of $\geq$60 U/mL was considered positive. The TgAb level was also measured by radioimmunoassay (BRAHMS anti-Tgn RIA), and a value of $\geq$ 60 U/mL was considered positive [19].

## Statistical analysis

Statistical analyses were performed using the R program (version 3.5.1, R Foundation for Statistical Computing, Vienna, Austria; http://www.R-project.org). Continuous variables are presented as median and Inter Quartile Range (IQR), and categorical variables are presented as numbers (percentages).

# Results

## Baseline characteristics of patients

A total of 11043 patients underwent thyroid lobectomy for thyroid cancer between 2010 and 2019. Among them, 26 (0.2%) were newly diagnosed with GD during follow-up (Fig 1). The baseline clinical and pathological characteristics of the 26 patients are described in Table 1. The median age was 43.8 years (IQR 34.1–44.4), and 88.5% were female. 3.8% and 7.7% of the patients were current and ex-smokers, respectively. All patients were in the euthyroid state before thyroid surgery with a median TSH level of 2.3 μM/mL and fT4 level of 1.2 ng/dL. Preoperative thyroid autoantibody tests were performed in 18 (69.2%) patients, and TPOAb and TgAb positivity were detected in 11 (61.1%) and 8 (44.4%) patients, respectively. None of the

**Table 1. Baseline characteristics of patients with Grave's disease after lobectomy.**

|  | Total n = 26 |
|---|---|
| **Age (years)** | 43.8 (34.1–44.4) |
| < 55 years | 20 (76.9%) |
| **Sex** |  |
| Female | 23 (88.5%) |
| **Smoking** |  |
| Current smoker | 1 (3.8%) |
| Ex-smoker | 2 (7.7%) |
| **Goiter (yes)** | 0 (0%) |
| **Thyroid function test** |  |
| TSH (μM/mL) (ref 0.4–5.0) | 2.3 (1.0–2.4) |
| Free T4 (ng/dL) (ref 0.8–1.9) | 1.2 (1.2–1.3) |
| **Thyroid autoantibody before lobectomy** |  |
| TPOAb [a] (positive) | 11 (61.1%) |
| TgAb [a] (positive) | 8 (44.4%) |
| **TBII [b]** | Not checked |
| **Pathologic type of PTC** |  |
| Classical type PTC | 23 (88.5%) |
| Follicular variant PTC | 3 (11.5%) |
| **Tumor size (cm)** | 0.7 (0.5–0.9) |
| ≤ 1cm | 18 (69.2%) |
| **Lymph node metastasis (N1a)** | 6 (23.1%) |
| **Multifocality** | 6 (23.1%) |
| **Lymphocytic thyroiditis** | 16 (61.5%) |

Continuous variables are presented as median (interquartile range) and categorical variables as numbers (percentage)

[a]TPOAb and TgAb were measured only in 18 patients.

[b] None of the patients were checked for TBII before lobectomy

TSH, thyroid stimulating hormone; TPOAb, anti-thyroid peroxidase antibody; TBII, thyrotropin binding inhibitor immunoglobulin; TgAb, antithyroglobulin antibody; PTC, papillary thyroid cancer.

patients had goiter before thyroid surgery or were checked for TBII. All patients were diagnosed with PTC, and the subtypes were as follows: classical type in 23 (88.5%) and follicular variant in 3 (11.5%) patients. The median tumor size was 0.7 cm (0.5–0.9) and 69.2% were PTMCs. Cervical lymph node metastasis was confirmed in 6 (23.1%) patients, and no structural recurrence was observed during a median 6 years of follow-up. Lymphocytic thyroiditis was found in 16 (61.5%) patients, and 11 (42.3%) patients were on levothyroxine after lobectomy due to confirmed hypothyroidism.

## Diagnosis of GD

The median duration between thyroid surgery and diagnosis of GD was 3.3 (IQR 2.3–4.9) years; however, the time interval was very variable in different patients (Fig 2). Among 26 patients, 13 (50%) were diagnosed during routine follow-up of the thyroid function tests and had no definite symptoms (Table 2 and Fig 2). Ten (38.5%) patients had an unplanned visit due to symptoms of thyrotoxicosis such as tremors, palpitations, weight loss, and diarrhea. GD

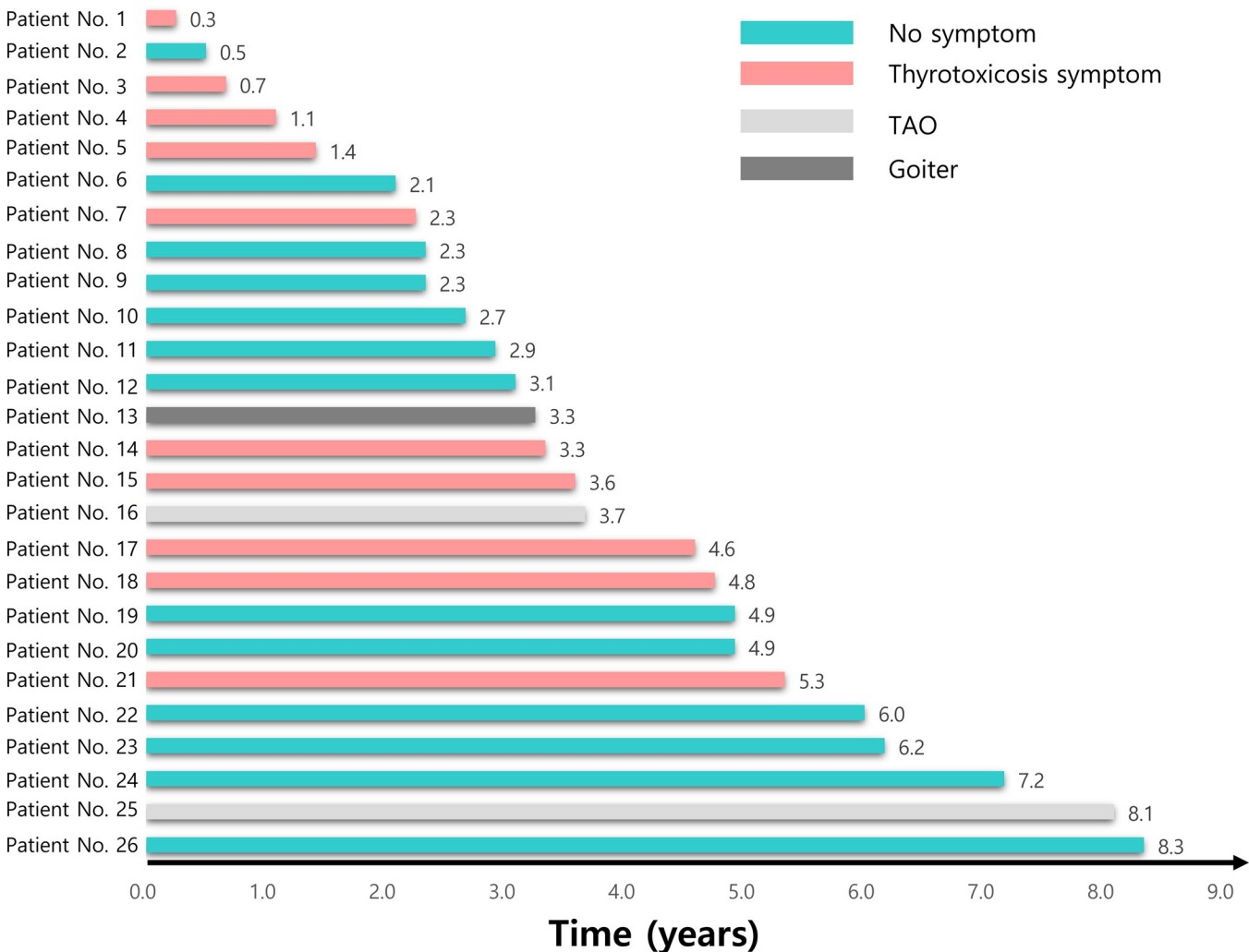

**Fig 2. Diagnosis of Graves' disease.** This figure shows the individual data of the 26 patients according to the time interval from thyroid lobectomy until the diagnosis of Graves' disease and the associated symptoms of Graves' disease. TAO, thyroid associated orbitopathy.

**Table 2. Characteristics of patients who developed Graves' disease after thyroid lobectomy.**

| | Total n = 26 |
|---|---|
| **Diagnosis reason** | |
| Routine follow-up of thyroid function tests | 13 (50%) |
| **Thyrotoxicosis symptoms [a]** | 10 (38.5%) |
| Development of goiter | 1 (3.8%) |
| Development of TAO | 2 (7.7%) |
| **Patients who have previously taken levothyroxine** | 11 (42.3%) |
| **Thyroid autoantibody at diagnosis** | |
| TPOAb (positive) | 20 (76.9%) |
| TgAb (positive) | 14 (53.8%) |
| TBII (positive) | 26 (100%) |
| **Thyroid scan [b]** | |
| **Increased uptake rate in remnant thyroid lobe** | 15 (100%) |
| **TAO (yes)** | 3 (11.5%) |
| **Graves' disease treatment** | |
| Antithyroid drug | 21 (80.8%) |
| RAI therapy | 2 (7.7%) |
| Completion thyroidectomy | 3 (11.5%) |
| **Graves' disease state in patients taking ATDs** | |
| Remission after discontinuation of ATD | 11 (52.4%) |
| On ATD | 10 (47.6%) |

Continuous variables are presented as median (interquartile range) and categorical variables as numbers (percentage).

[a] Thyrotoxicosis symptoms include tremors, palpitations, weight loss, and diarrhea.

[b] Thyroid scan was performed in 15 patients.

TAO, thyroid-associated orbitopathy; TPOAb, anti-thyroid peroxidase antibody; TgAb, antithyroglobulin antibody; TBII, thyrotropin binding inhibitor immunoglobulin; RAI, radioactive iodine; ATD, antithyroid drug.

was diagnosed later in one patient who had newly developed goiter in the remnant thyroid lobe and in two referral patients who were diagnosed with TAO in the ophthalmology unit. One more patient who initially had no symptoms of thyrotoxicosis was also diagnosed with TAO during the treatment for GD. All patients underwent the autoantibody test when they were diagnosed with GD, and the positivity of TPOAb and TgAb was 76.9% and 53.8%, respectively, which were higher than the preoperative positivity rates. Among patients in whom the autoantibodies were measured preoperatively, 2 and 3 patients respectively showed positive conversion of TPOAb and TgAb, at the time of GD diagnosis. All the patients showed positive TBII at the time of diagnosis of GD.

## Treatment and course of GD

For the treatment of GD, all patients received ATDs as the first-line therapy, and 21 (80.8%) continued ATDs for maintenance. RAI therapy and complete thyroid surgery were performed in 2 (7.7%) and 3 (11.5%) patients, respectively. All these five patients were on levothyroxine replacement after RAI therapy or surgery. Among 21 patients who were maintained on ATDs, 11 (52.4%) discontinued ATDs after a median 16 months of treatment and had remission of GD. However, one patient had a relapse of GD after 20 months of remission, and 10 (47.6%) patients were on maintenance ATDs at the last follow-up.

## Discussion

We found that the incidence of GD after lobectomy for thyroid cancer is 0.2% in a large cohort. Among patient who diagnosed with GD, 61.1% and 44.4% were positive for TPOAb and TgAb, respectively, before surgery. These rates are higher than those seen in the general population of patients with thyroid cancer, considering that autoantibody positivity is reported as approximately 18–23% in such patients [19–21]. Furthermore, lymphocytic thyroiditis is documented in 61.5% of surgical specimens. These findings indicate that patients who developed GD were more likely to have autoimmune thyroid disease compared with those who did not develop GD. However, preoperatively confirmed autoimmune thyroid disease alone cannot be used as a predictor of the occurrence of GD after surgery, and we could not find other relevant preoperative predictive factors.

In previous studies, the incidence of GD diagnosed after thyroid surgery has been reported to be 0.08–0.24%, which is consistent with that in our study [13, 22]. Considering that the median time of 3.3 years that GD occurrence after surgery in this study, patients who underwent thyroid surgery between 2018 and 2019 may have a short follow-up interval to disease occurrence. However, when we assessed GD incidence excluding the patients who underwent lobectomy between 2018 and 2019, the incidence (0.29%) was similar to that of the total cohort.

Similarly, Kasuga et al. also emphasized the presence of autoimmune thyroiditis in patients who developed GD [13]. They presented a case series of patients with GD and stated that the incidence of postoperative GD in patients with positive autoantibody (1.5%) was significantly higher than that (0.12%) in the autoantibody negative group [13], which is also consistent with our results. Our study and previous studies suggested that preoperative measurement of autoantibodies might be helpful in predicting the occurrence of GD in patients undergoing lobectomy. However, the positive conversion of TPOAb and TgAb after the development of GD was seen in several patients in this study, and the development of GD among patients with autoimmune thyroid disease was not common. Thus, prospective studies with large sample sizes are necessary to verify the clinical implications.

The clinical course of GD that occurred after lobectomy in this study was similar to that in other patients with GD. In this study, 81% of the patients received ATDs as a definite long-term treatment, and 19% received ATDs as a bridging therapy until RAI or surgery. Among patients who received ATDs, approximately 52% of the patients had a remission, and others were still taking ATDs during a median follow-up period of 1.8 years. The remission rate was similar to that reported in patients with GD reported in Europe and Japan [23–25]. However, the results for TAO were different from those reported in the previous studies. In general, approximately 5–6.1% of patients with GD have moderate too severe TAO [3, 18]. However, TAO was observed in 3 (11.5%) patients in the present study, and all of them had a severe disease state that required steroid pulse therapy.

GD is a complex disease with autoimmune pathophysiology, which results from the interactions between genetic and environmental factors [2, 26, 27]. The pathogenesis of the development of GD after lobectomy is unclear; however, some hypotheses exist. First, the abnormality in antigen-presenting cells that sustain the activation of suppressor and regulatory cells, which then attack the immune system and cause GD [28]. The second hypothesis is that mechanical and biochemical stress from surgery causes neuroendocrine fluctuations, which affect immunological homeostasis [29, 30]. The time to diagnosis of GD after lobectomy ranged from 0.3 to 8.2 years in this study. The variance in the time until onset of the disease was also seen in previous studies, which reported a range of 9 months to 27 years [12–14]. This might be related to the complex pathogenesis of GD, making it difficult to predict the occurrence of disease

after lobectomy. Three patients were diagnosed with GD within 1 year after lobectomy in this study. All of them got surgery due to incidentally found thyroid nodules, and there were no clinical findings that suggest GD preoperatively. Furthermore, all three patients showed a euthyroid state in the preoperative thyroid function test. However, No.1 patient (Fig 2) diagnosed with GD at her first follow-up visit, it is impossible to rule out the possibility of GD undetected before the lobectomy, or surgery itself might act as a stress factor to cause GD.

This study has some limitations. First, it was a retrospective study from a single tertiary center which makes it difficult to generalize the findings. Second, we could not directly compare patients who developed with GD to those who did not develop GD because the incidence was too low. However, thyroid autoimmunity was the most important characteristic of patients who developed GD after lobectomy in this study. This is obviously higher than that seen in the general population with thyroid cancer, and a direct comparison between the two groups might not be necessary. The strength of this study is the large sample size to evaluate the incidence and clinical course of GD diagnosed after lobectomy for thyroid cancer.

## Conclusion

In conclusion, although rare, GD can occur in remnant thyroid after lobectomy. Thus, surgeons should consider the possibility of GD when hyperthyroidism is found during the follow-up of patients after lobectomy, and an appropriate workup is required. Preoperative measurements of autoantibodies might be helpful to predict the occurrence of GD; however, more evidence for the same is required.

## Acknowledgments

We would like to thank Editage (www.editage.co.kr) for English language editing.

## Author Contributions

**Conceptualization:** Min Ji Jeon.

**Data curation:** Ahreum Jang, Won Gu Kim, Tae Yong Kim, Won Bae Kim, Young Kee Shong.

**Formal analysis:** Meihua Jin.

**Writing – original draft:** Meihua Jin.

**Writing – review & editing:** Min Ji Jeon.

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
