## [Decision Letter · Decision Letter 0]

13 Dec 2021

PONE-D-21-23185

Graves’ Disease Diagnosed in Remnant Thyroid After Lobectomy for Thyroid Cancer

PLOS ONE

Dear Dr. Jeon,

Thank you for submitting your manuscript to PLOS ONE. After careful consideration, we feel that it has merit but does not fully meet PLOS ONE’s publication criteria as it currently stands. Therefore, we invite you to submit a revised version of the manuscript that addresses the points raised during the review process.

We look forward to receiving your revised manuscript.

Kind regards,

Byeong-Cheol Ahn, M.D., Ph.D.

Academic Editor

PLOS ONE

Journal Requirements:

"This study was supported by a grant (2021IL0025) from the Asan Institute for Life Sciences, Asan Medical Center, Seoul, Korea."

"This study was supported by a grant (2021IL0025) from the Asan Institute for Life Sciences, Asan Medical Center, Seoul, Korea."

Reviewers' comments:

Reviewer's Responses to Questions

**Comments to the Author**

1. Is the manuscript technically sound, and do the data support the conclusions?

Reviewer #1: No

Reviewer #2: Yes

Reviewer #3: Yes

2. Has the statistical analysis been performed appropriately and rigorously? 

Reviewer #1: No

Reviewer #2: N/A

Reviewer #3: Yes

3. Have the authors made all data underlying the findings in their manuscript fully available?

Reviewer #1: Yes

Reviewer #2: Yes

Reviewer #3: Yes

4. Is the manuscript presented in an intelligible fashion and written in standard English?

Reviewer #1: No

Reviewer #2: Yes

Reviewer #3: Yes

5. Review Comments to the Author

Reviewer #1: Jin et al. have demonstrated the prevalence of Graves’ disease after lobectomy in the patients with thyroid cancer. They have also showed high positive rate of anti-TPO and Tg antibodies at preoperative period in the patients with Graves’ disease developed in remnant thyroid after lobectomy for thyroid cancer.

As the authors have not performed any statistical analyses to characterize the cases with Graves’ disease after thyroid lobectomy, the data presentation is descriptive and the discussion is weak.

To begin with, the authors should demonstrate not only the features of 26 Graves’ cases, but also those of 11017 cases without Graves’ disease in Table1. Then, the authors should simply perform Chi-squared or Fisher’s exact test to explore the factors which is different between the cases with and without Graves’ disease.

It is not surprising that high prevalence of Graves’ disease is observed in the cases with positive Tg or TPO antibodies.

It is clinically important to clarify whether thyroid lobectomy itself may increase the prevalence of autoimmune thyroid disorders including Graves’ disease. To approach the issue, the authors should provide another data set of age-sex-matched control cases such as benign nodules without operation to perform a case-control study.

The contribution of this study to the existing knowledge is insufficient.

Reviewer #2: This study assessed the incidence and clinicopathological characteristic of newly developed Graves’ disease after lobectomy for thyroid cancer. Although it is a single-center retrospective study, the results of analysis using 10 years of data are thought to be a good reference for future prospective studies.

This is a well written paper with good syntax and makes an easy reading. And results are presented in a clear manner and easy to understand.

However, I would like to address a few Minor issues.

1. This study was performed on those who underwent thyroidectomy between 2010 and 2019. As it took median 3.3 years (IQR 2.3-4.9) to diagnose GD after surgery, the study interval seems to be short for those who operated 2018-2019 to track whether GD occurred. It seems that the incidence was rather lowered by this, so please give me your opinion.

2. 3 out of 26 patients were diagnosed with GD within 1 year after surgery. Are these cases among 8 patients who did not check preoperative TPO Ab and Tg Ab? I would like to ask for your opinion on whether the preexisting GD was undetected preoperatively.

3. In Table 1, please add the related information about ‘None of the patients had goiter before thyroid surgery or were checked for TBII’.

4. In Table 2, please add a detailed description of ‘Hyperthyroidism symptom’ and the thyroid scan findings.

Reviewer #3: The authors have assessed the incidence and clinicopathological characteristic of newly developed Graves’ disease after lobectomy for thyroid cancer. The manuscript is well written, and the data and the interpretation are technically sound and solid. Nevertheless, minor issues are raised to enhance the strength of the manuscript. As the authors have pointed out, this is not a comparison study between patients with or without Graves’ disease; therefore, the scientific importance is substantially limited.

1. The legend of Figure 2 does not properly explain the corresponding data. It seems that this is not the “distribution” but “individual data per se” of 26 patients with Graves’ disease.

2. In addition, the authors need to include each patient No. in left column of the graph in Figure 2.

3. Please check large alphabets are truly required in the legends in Table 1. Some full names start with large letters whereas the others are not.

eg. TSH, Thyroid Stimulating Hormone; TPOAb, Anti-Thyroid Peroxidase Antibody; TgAb, Antithyroglobulin Antibody; PTC, papillary thyroid cancer

4. Title of Table 2. Characteristics of Patients who Developed Grave's Disease after Thyroid Lobectomy

=> Grave’ disease is a typo of Graves’ disease

5. Table 2. Before LT4 treatment (yes) -> This requires English editing.

6. In general, tables not easy to follow. Please consider revising the format to enhance the quality.

6. PLOS authors have the option to publish the peer review history of their article (what does this mean?). If published, this will include your full peer review and any attached files.

Reviewer #1: No

Reviewer #2: No

Reviewer #3: No

---

## [Author Response · Author response to Decision Letter 0]

21 Dec 2021

Response to Reviewer 1

We appreciate your review of our manuscript. 

Reviewer #1: Jin et al. have demonstrated the prevalence of Graves’ disease after lobectomy in the patients with thyroid cancer. They have also showed high positive rate of anti-TPO and Tg antibodies at preoperative period in the patients with Graves’ disease developed in remnant thyroid after lobectomy for thyroid cancer. 

As the authors have not performed any statistical analyses to characterize the cases with Graves’ disease after thyroid lobectomy, the data presentation is descriptive and the discussion is weak. To begin with, the authors should demonstrate not only the features of 26 Graves’ cases, but also those of 11017 cases without Graves’ disease in Table1. Then, the authors should simply perform Chi-squared or Fisher’s exact test to explore the factors which is different between the cases with and without Graves’ disease. It is not surprising that high prevalence of Graves’ disease is observed in the cases with positive Tg or TPO antibodies. It is clinically important to clarify whether thyroid lobectomy itself may increase the prevalence of autoimmune thyroid disorders including Graves’ disease. To approach the issue, the authors should provide another data set of age-sex-matched control cases such as benign nodules without operation to perform a case-control study. The contribution of this study to the existing knowledge is insufficient.

→ Thank you for your comment, and we agree that our inability to directly compare patients who developed with GD to those who did not is the major limitation of this study. We have added this point as a limitation in the discussion part (Page 12 Line 215-217). However, as the primary purpose of the current study was to assess the incidence of GD after lobectomy and their clinical course, we think this study so far has been sufficient to achieve our purpose. In addition, regarding your comment, “if the thyroid lobectomy itself increases the prevalence of autoimmune thyroid disorder?” we look forward to implementing it in our future studies. 

Response to Reviewer 2

We appreciate your review of our manuscript. We believe that your comments have helped us to improve our manuscript. In the revised manuscript, changes are shown in bold red text. 

Reviewer #2: This study assessed the incidence and clinicopathological characteristic of newly developed Graves’ disease after lobectomy for thyroid cancer. Although it is a single-center retrospective study, the results of analysis using 10 years of data are thought to be a good reference for future prospective studies.

This is a well written paper with good syntax and makes an easy reading. And results are presented in a clear manner and easy to understand.

However, I would like to address a few Minor issues.

1. This study was performed on those who underwent thyroidectomy between 2010 and 2019. As it took median 3.3 years (IQR 2.3-4.9) to diagnose GD after surgery, the study interval seems to be short for those who operated 2018-2019 to track whether GD occurred. It seems that the incidence was rather lowered by this, so please give me your opinion.

→ Thank you for important comment. We agree with your opinion and assessed the incidence after excluding the patients who underwent lobectomy between 2018 and 2019. The incidence was similar to that of the total cohort. We have added this point in the discussion part (Page 10, Line 171–176).

2. 3 out of 26 patients were diagnosed with GD within 1 year after surgery. Are these cases among 8 patients who did not check preoperative TPO Ab and Tg Ab? I would like to ask for your opinion on whether the preexisting GD was undetected preoperatively.

→ Among the 3 patients, 2 patients did not check preoperative autoantibody, and one patient had positive autoantibody. All of them got surgery due to incidentally found thyroid nodules, and there were no clinical findings that suggest GD preoperatively. Furthermore, all three patients showed a euthyroid state in the preoperative thyroid function test. However, No.1 patient (Fig2) diagnosed with GD at her first follow-up visit, it is impossible to rule out the possibility of GD undetected before the lobectomy, or surgery itself might act as a stress factor to cause GD. We have added this point in discussion (Page 12, Line 207-213). 

3. In Table 1, please add the related information about ‘None of the patients had goiter before thyroid surgery or were checked for TBII’.

→ Thank you for your comment. We have added the description of goiter and TBII in Table 1. 

In addition, we have described the definition of goiter in method part (Page6, Line 83–85)

4. In Table 2, please add a detailed description of ‘Hyperthyroidism symptom’ and the thyroid scan findings.

→ Thank you for your comment. We have added the description of hyperthyroidism symptoms in footnote of Table 2. As most of the patients complained of multiple overlapping thyrotoxicosis symptoms, it is difficult to subclassifying them in Table 2. 

Response to Reviewer 3

We appreciate your review of our manuscript. We believe that your comments have helped us to improve our manuscript. In the revised manuscript, changes are shown in bold red text.

Reviewer #3: The authors have assessed the incidence and clinicopathological characteristic of newly developed Graves’ disease after lobectomy for thyroid cancer. The manuscript is well written, and the data and the interpretation are technically sound and solid. Nevertheless, minor issues are raised to enhance the strength of the manuscript. As the authors have pointed out, this is not a comparison study between patients with or without Graves’ disease; therefore, the scientific importance is substantially limited.

1. The legend of Figure 2 does not properly explain the corresponding data. It seems that this is not the “distribution” but “individual data per se” of 26 patients with Graves’ disease.

→ Thank you for your comment. We agree with your opinion and have revised the legend of Fig 2 (Page 9, Line 147–148)

2. In addition, the authors need to include each patient No. in left column of the graph in Figure 2.

→ We have revised the figure 2 as your suggestion. 

3. Please check large alphabets are truly required in the legends in Table 1. Some full names start with large letters whereas the others are not.

eg. TSH, Thyroid Stimulating Hormone; TPOAb, Anti-Thyroid Peroxidase Antibody; TgAb, Antithyroglobulin Antibody; PTC, papillary thyroid cancer

→ Legends of tables are revised as your suggestion. Sorry for your inconvenience. 

4. Title of Table 2. Characteristics of Patients who Developed Grave's Disease after Thyroid Lobectomy

=> Grave’ disease is a typo of Graves’ disease

→ We have corrected the typo in Table 2. Sorry for your inconvenience. 

5. Table 2. Before LT4 treatment (yes) -> This requires English editing.

→ Thank you for your comment. We have revised it as “Patients who have previously taken levothyroxine” in Table 2. 

6. In general, tables not easy to follow. Please consider revising the format to enhance the quality.

→ We have revised the table1 format in accordance with your suggestion (Table 1 and 2).

---

## [Decision Letter · Decision Letter 1]

1 Mar 2022

Graves’ Disease Diagnosed in Remnant Thyroid After Lobectomy for Thyroid Cancer

PONE-D-21-23185R1

Dear Dr. Jeon,

We’re pleased to inform you that your manuscript has been judged scientifically suitable for publication and will be formally accepted for publication once it meets all outstanding technical requirements.

Kind regards,

Byeong-Cheol Ahn, M.D., Ph.D.

Academic Editor

PLOS ONE

Additional Editor Comments (optional):

Reviewers' comments:

Reviewer's Responses to Questions

**Comments to the Author**

1. If the authors have adequately addressed your comments raised in a previous round of review and you feel that this manuscript is now acceptable for publication, you may indicate that here to bypass the “Comments to the Author” section, enter your conflict of interest statement in the “Confidential to Editor” section, and submit your "Accept" recommendation.

Reviewer #1: (No Response)

Reviewer #2: All comments have been addressed

Reviewer #3: All comments have been addressed

2. Is the manuscript technically sound, and do the data support the conclusions?

Reviewer #1: No

Reviewer #2: Yes

Reviewer #3: Yes

3. Has the statistical analysis been performed appropriately and rigorously? 

Reviewer #1: No

Reviewer #2: N/A

Reviewer #3: Yes

4. Have the authors made all data underlying the findings in their manuscript fully available?

Reviewer #1: No

Reviewer #2: Yes

Reviewer #3: Yes

5. Is the manuscript presented in an intelligible fashion and written in standard English?

Reviewer #1: No

Reviewer #2: Yes

Reviewer #3: Yes

6. Review Comments to the Author

Reviewer #2: This study assessed the incidence and clinicopathological characteristic of newly developed Graves’ disease after lobectomy for thyroid cancer.

The authors have adequately addressed the comments raised in a previous round of review.

Reviewer #3: The authors have faithfully addressed this reviewer's concerns. Therefore I do not have further comments.

7. PLOS authors have the option to publish the peer review history of their article (what does this mean?). If published, this will include your full peer review and any attached files.

Reviewer #1: No

Reviewer #2: No

Reviewer #3: No

---

## [Editor Report · Acceptance letter]

3 Mar 2022

PONE-D-21-23185R1 

Graves’ disease diagnosed in remnant thyroid after lobectomy for thyroid cancer 

Dear Dr. Jeon:

I'm pleased to inform you that your manuscript has been deemed suitable for publication in PLOS ONE. Congratulations! Your manuscript is now with our production department. 

Kind regards, 

on behalf of

Professor Byeong-Cheol Ahn 

Academic Editor

PLOS ONE